# Natural History of Metabolic Dysfunction-Associated Steatotic Liver Disease: From Metabolic Syndrome to Hepatocellular Carcinoma

**DOI:** 10.3390/medicina61010088

**Published:** 2025-01-07

**Authors:** Melchor Alpízar Salazar, Samantha Estefanía Olguín Reyes, Andrea Medina Estévez, Julieta Alejandra Saturno Lobos, Jesús Manuel De Aldecoa Castillo, Juan Carlos Carrera Aguas, Samary Alaniz Monreal, José Antonio Navarro Rodríguez, Dulce María Fernanda Alpízar Sánchez

**Affiliations:** 1Endocrinology, Specialized Center for Diabetes, Obesity and Prevention of Cardiovascular Diseases (CEDOPEC), Mexico City 11650, Mexico; 2Clinical Research, Specialized Center for Diabetes, Obesity and Prevention of Cardiovascular Diseases (CEDOPEC), Mexico City 11650, Mexico; reizzlesam@gmail.com (S.E.O.R.); andy.9525@hotmail.com (A.M.E.); julietaasl97@gmail.com (J.A.S.L.); salaniz@anahuac.com (S.A.M.); a01335659@gmail.com (J.A.N.R.); feralpizar@gmail.com (D.M.F.A.S.); 3Clinical Nutrition, Specialized Center for Diabetes, Obesity and Prevention of Cardiovascular Diseases (CEDOPEC), Mexico City 11650, Mexico; jccxa12345@gmail.com

**Keywords:** Metabolic Dysfunction-Associated Steatotic Liver Disease, steatohepatitis, liver cirrhosis, carcinoma hepatocellular, elastography

## Abstract

*Introduction*: Metabolic Dysfunction-Associated Steatotic Liver Disease (MASLD) stems from disrupted lipid metabolism in the liver, often linked to obesity, type 2 diabetes, and dyslipidemia. In Mexico, where obesity affects 36.9% of adults, MASLD prevalence has risen, especially with metabolic syndrome affecting 56.31% by 2018. MASLD can progress to Metabolic Dysfunction-Associated Steatohepatitis (MASH), affecting 5.27% globally, leading to severe complications like cirrhosis and hepatocellular carcinoma. *Background*: Visceral fat distribution varies by gender, impacting MASLD development due to hormonal influences. Insulin resistance plays a central role in MASLD pathogenesis, exacerbated by high-fat diets and specific fatty acids, leading to hepatic steatosis. Lipotoxicity from saturated fatty acids further damages hepatocytes, triggering inflammation and fibrosis progression in MASH. Diagnosing MASLD traditionally involves invasive liver biopsy, but non-invasive methods like ultrasound and transient elastography are preferred due to their safety and availability. These methods detect liver steatosis and fibrosis with reasonable accuracy, offering alternatives to biopsy despite varying sensitivity and specificity. *Conclusions*: MASLD as a metabolic disorder underscores its impact on public health, necessitating improved awareness and early management strategies to mitigate its progression to severe liver diseases.

## 1. Introduction

The liver plays a crucial role in the lipid metabolism. It acts as the central regulator of lipid homeostasis by overseeing the synthesis, export, and redistribution of new fatty acids for their usage across a variety of organs and tissues that need them [1].

Hepatic fat accumulation, better known as hepatic steatosis, has a long variety of etiologies: lifestyle, alcohol consumption, and viral infections, among others. There are also multiple risk factors that are collectively responsible for the onset and progression of non-alcoholic fatty liver disease (NAFLD). The most well-known factors are obesity, type 2 diabetes mellitus (T2DM), and dyslipidemia [2,3,4,5].

Despite the high prevalence of the previously mentioned risk factors, the World Health Organization (WHO) has yet to recognize NAFLD as a non-communicable disease of priority [6].

Recently, an international consensus of experts proposed the redefinition of NAFLD as Metabolic Dysfunction-Associated Steatotic Liver Disease (MASLD) [7,8]. This new definition includes metabolic parameters as diagnostic criteria for patients with NAFLD. This change is better explained because it was commonly believed that only individuals with elevated BMI are at a bigger risk of developing NAFLD. However, many individuals with normal weight also develop the condition. Regardless of this, all affected individuals tend to exhibit metabolic and clinical alterations similar to those seen in obesity and metabolic syndrome (MetS) [9,10,11]. There are two etiologies that define hepatic steatosis in different ways. When it develops due to an inadequate lifestyle without alcohol abuse, it is referred to as MASLD. On the other hand, when it develops due to the abuse of alcohol alone (140–350 g/wk for females and 210–420 g/wk for males), it is called alcohol-associated/related liver disease (ALD). When it is a combination of alcohol consumption and lipotoxicity due to an inadequate lifestyle, it is referred to as metabolic and alcohol-related/associated liver disease (MetALD) [7]. For the purposes of this research, the new terminology will be used even though it refers to studies published prior to this one.

## 2. Epidemiology of MASLD and Its Stages in the Adult Mexican Population and Globally

In Mexico, the prevalence of obesity among adults is 36.9%, and overweight affects 38.3% of the population. Women exhibit a 45% higher likelihood of developing obesity compared to men [12], underscoring the elevated risk in this population for developing comorbidities associated with high mortality and diminished quality of life.

Metabolic syndrome represents a critical factor significantly elevating the risk of developing MASLD in affected men and women (OR 4.00 and OR 11.20, respectively) [13]. In Mexico, the prevalence of metabolic syndrome has shown an increasing trend. The increase becomes more evident when contrasting the prevalence of this condition, which was 40.25% in 2006, sharply rising to 56.31% by 2018 [14].

Talking about MASLD in Mexico, by 2006, the prevalence was about 14% [15]. However, by 2018, this had an increment that went to 49.9%, highlighting its strong correlation with obesity, as only 7.42% of participants exhibited a normal BMI [16]. Globally, MASLD affects 32.4% of the population, with the proportion of total deaths attributable to MASLD rising from 0.10% to 0.17% [6,17]. Disability-adjusted life years (DALYs) related to high BMI have seen a rise of 12.7% for women and 26.8% for men [6]. Notably, a significant proportion of lean and non-obese individuals also present with MASLD, with prevalences of 19.2% and 40.8%, respectively, within these populations [10]. The global prevalence of MASLD is estimated at approximately 38.77% [18].

MASLD progresses to Metabolic Dysfunction-Associated Steatohepatitis (MASH), with a global prevalence estimated at 5.27% and the highest prevalence found in Latin America at 7.11% [19].

Following progression from MASH, patients may develop cirrhosis. Globally, the prevalence of compensated and decompensated cirrhosis has increased by 33.2% and 54.8%, respectively, representing a significant rise in cases [20]. Recent evidence indicates that MASLD has transitioned from being the third leading cause of cirrhosis, responsible for 14% of incident cases in 2000, to becoming the predominant cause since 2012, contributing to 36% of incident cases in 2019 [21].

The natural progression of hepatic cirrhosis culminates in hepatocellular carcinoma (HCC), which remains globally the fourth-leading cause of cancer-related mortality, following lung, breast, and colorectal cancers. HCC can manifest in the absence of cirrhosis [22], and its prevalence is attributed to hepatitis B in 40% of cases, hepatitis C in another 40%, alcohol consumption in 11%, and approximately 10% due to other factors such as MASLD. However, these proportions are expected to undergo significant shifts due to the increasing prevalence of cirrhosis associated with MASH [17,22]. Notably, MASH is currently the fastest-growing cause of age-adjusted liver cancer deaths globally, contrasting with the decline of age-adjusted deaths from hepatitis B and C [22].

## 3. Etiology: From Visceral Fat to Liver Steatosis

### 3.1. Visceral Fat Anatomy

Visceral fat is categorized into intraperitoneal fat, which includes greater omental fat, lesser omental fat, mesenteric fat, and mesocolonic fat. This is classified as adipose tissue itself, composed of triglycerides and free fatty acids that contribute to lipotoxicity, unlike fat surrounding retroperitoneal organs. Lymphatic circulation, in this location, begins with smaller lymphatic vessels within the mesenteries, subsequently progressing into larger lymphatic vessels located in the retroperitoneal area. Therefore, both the mesenteries and retroperitoneum receive the same supply of lipid-rich chyle [23].

### 3.2. Development of Visceral Adiposity

There is a clear distinction in visceral fat accumulation between men and women, characterized by android morphology in men (greater visceral fat) and gynoid morphology in women (greater subcutaneous fat). Various mechanisms contribute to this difference, involving hormonal, anatomical, and physiological [23].

#### 3.2.1. Sexual Hormones

Estrogens play a key role in premenopausal women by reducing lipolysis in adipocytes of subcutaneous adipose tissue through inhibition of hormone-sensitive lipase (HSL) [24]. Women have a higher number of fat depots in subcutaneous tissue [25] and greater activity of lipoprotein lipase (LPL), enhancing their capacity for fat storage in this area. Additionally, women exhibit a higher catabolic rate of hepatic-derived lipoproteins (triglyceride-rich, very low-density lipoproteins), facilitating a more effective redistribution of fat storage from the liver to subcutaneous fat compared to men [26]. Conversely, testosterone suppresses LPL activity in subcutaneous adipose tissue [25]. Studies confirm that in men, 21% of ingested fat is stored as intraperitoneal fat [23,27], whereas in women, this figure is only around 5% [23,28].

#### 3.2.2. Insulin Resistance

Hepatic steatosis is strongly associated with adherence to a Western diet pattern compared to other dietary patterns (OR: 2.04, *p* < 0.001) [29]. Therefore, a thorough exploration of the pathophysiological mechanisms leading to its development is crucial, notably insulin resistance [9,30], which plays a pivotal role in the onset of MASLD. This mechanism has been corroborated in C57BL/6J murine models [31], underscoring the intimate relationship between these pathological conditions.

The decreased sensitivity to insulin also impacts lipid metabolism, leading to various metabolic abnormalities. However, first and foremost, one must understand the physiological function of insulin: In peripheral tissues, postprandial insulin triggers the translocation of GLUT4, initiating a lipogenic effect by suppressing lipolysis, primarily via HSL in adipocytes. Upon binding to its cellular receptor, insulin activates it, leading to subsequent tyrosine phosphorylation of the insulin receptor substrate (IRS) and activation of phosphatidylinositol-3-kinase (PI3K) [32]. Additionally, insulin inhibits gluconeogenesis by suppressing phosphoenolpyruvate carboxykinase and glucose-6-phosphatase while promoting glycogen synthesis through glycogen synthase phosphorylation, thereby regulating blood glucose levels. Glucose undergoes conversion into acetyl-CoA, with excess utilized in the de novo lipogenesis (DNL). Concurrently, acetyl-CoA carboxylase catalyzes the conversion of acetyl-CoA to malonyl-CoA, inhibiting fatty acid β-oxidation and further promoting lipogenesis [33,34].

The defects of insulin signaling and its peripheral resistance have been related to the accumulation of intra-myocellular and intra-hepatocellular lipids [35]. There is a record that the insulin signaling suppression induced by lipids through IRS-1 and IRS-2 is responsible for peripheral insulin resistance [36,37].

The pathophysiological connection among free fatty acids, hepatic steatosis, and insulin resistance has been firmly established through various animal models. High-fat diets (HFDs) have been demonstrated to induce MASLD and MASH, as well as hepatic and peripheral insulin resistance [35]. Roden et al. were pioneers in demonstrating that elevated levels of free fatty acids in humans can diminish insulin sensitivity [38]. Importantly, saturated fatty acids (SFAs) have been identified as key contributors to hepatic lipogenic dysregulation, distinct from the effects of mono- or polyunsaturated fats [39,40,41].

#### 3.2.3. Hepatic Fat Accumulation Pathways

Hepatic fat accumulation results from an imbalance between lipid acquisition and disposal, regulated through four main pathways: uptake of circulating lipids, DNL, fatty acid oxidation, and very low-density lipoprotein (VLDL) export.

Upon uptake by hepatocytes, fatty acids, being hydrophobic, do not diffuse freely through the cytosol. Instead, they rely on specific fatty acid-binding proteins (FABPs) for transport, primarily FABP1, which predominates in the liver. FABP1 facilitates the transport, storage, and utilization of fatty acids, thereby exerting a protective effect against lipotoxicity [42]. Additionally, FABP1 influences the expression of PPARα and PPARγ by mediating the transport of their ligands into the nuclei of hepatocytes [43]. Clinical studies have demonstrated overexpression of FABP1 in patients with obesity and hepatic steatosis compared to obese individuals without liver steatosis. Conversely, FABP1 levels are diminished in patients with MASH and mild fibrosis and completely suppressed in those with advanced fibrosis [44]. This suggests that increased FABP1 levels in the early stages of MASLD may enhance lipid flux as a compensatory mechanism to mitigate lipotoxicity. Conversely, declining FABP1 levels correspond with increased hepatic lipid accumulation, exacerbating lipotoxic damage to cellular organelles and essential hepatic cells.

As previously mentioned, DNL begins with acetyl-CoA. The synthesized fatty acids undergo desaturation, elongation, and esterification processes, ultimately leading to triglyceride formation, which can either be stored or exported via VLDL particles. The feedback loop inherent in this lipogenic process is implicated in the development of MASLD. Furthermore, individuals with MASLD exhibit increased DNL, thereby exacerbating hepatic steatosis [45,46]. This lipogenic state contributes approximately 26% of serum triglycerides and remains unimpeded in both pre- and postprandial states in MASLD patients [47].

Two transcription factors are in control of DNL: the sterol regulatory element-binding protein 1c (SREBP1c), which is activated by insulin, and the liver X receptor α. SREBP1c expression is markedly elevated in patients with MASLD, indirectly contributing to the development of insulin resistance by promoting lipogenesis. This enhancement of lipogenesis leads to the accumulation of harmful lipids, such as diacylglycerols, which can disrupt insulin signaling [48].

The reason why some individuals accumulate more hepatic fat compared to others during weight gain remains unknown.

## 4. Pathophysiology: Progression from MASLD to HCC

### 4.1. First Step: From MASLD to MASH

Lipotoxicity, resulting from the accumulation of SFAs in the liver cells, drives the progression of MASLD to MASH, leading to alterations in cellular organelles. In terms of mitochondrial changes, these include depolarization anomalies, release of cytochrome C, increased generation of reactive oxygen species (ROS), reduced mitochondrial biogenesis, disrupted autophagy, and decreased β-hydroxyacyl-CoA dehydrogenase activity, thereby impairing fatty acid β-oxidation [49,50]. Consequently, alternative pathways for fatty acid oxidation, such as ω-oxidation, become necessary [51]. ω-oxidation contributes to sustained inflammation in MASLD, as one of its steps, hydroxylation, is catalyzed by the enzymes CYP4A14 and CYP4A11, which are part of the CYP4A enzyme complex [52,53]. These enzymes have been linked to increased expression of TNFα, IL-1β, and IL-6, enhanced ROS synthesis, and the progression of hepatic injury [52,54]. Additionally, intracellular alterations include increased lysosomal permeability and cathepsin B redistribution [49]. The endoplasmic reticulum, as an adaptive response to lipotoxicity, exacerbates the condition by increasing autophagy, apoptosis, and inflammation in these patients [55,56].

Thus, lipotoxicity in patients with MASLD results in a persistent state of inflammation [57], exacerbated by the effects of insulin resistance. One of the molecular mechanisms by which insulin resistance contributes to the development of MASLD and its progression to MASH is the release of molecules that perpetuate low-grade inflammation, such as IL-6, CCL2/MCP-1, and CCL19 [58,59,60,61].

Another mechanism that has recently been investigated as an important factor in the development of inflammation in MASLD is the reduction in lysosomal acid lipase (LAL) levels [62]. This enzyme is responsible for hydrolyzing cholesterol esters and triglycerides into free fatty acids and unesterified cholesterol [63]. In MASLD, its reduction or deficiency has been observed to increase lipid accumulation, thereby contributing to lipotoxicity and the development of necroinflammation [64].

This necroinflammation is characterized by hepatocyte death and the dysregulated activation of Kupffer cells, which are macrophages that induce inflammation and fibrosis in MASH in both humans and murine models [65]. Another critical factor in lipotoxicity-induced hepatic injury is the activation of hepatic stellate cells (HSCs). During pathological liver conditions, HSCs are responsible for producing collagen, proteoglycans, and glycoproteins, acting as precursors to myofibroblasts and contributing to increased extracellular matrix deposition [66]. These cells are activated by the liver’s inflammatory response and autophagy induced by markers such as TGF-β1 [67]. The liberation of these inflammatory markers induces the production of the necessary energy for HSC activation through ATP synthesis via β-oxidation of fatty acids stored in hepatocyte cytoplasmic droplets, thereby promoting fibrosis [68]. Consequently, metabolic factors contribute to sustained inflammation, creating a vicious cycle that perpetuates liver damage and may lead to the development of fibrosis, cirrhosis, and hepatocellular carcinoma.

All the previously mentioned mechanisms contribute to the natural history of MASLD. MASLD is initially characterized by hepatic steatosis of ≥5%, which can advance to MASH when inflammation and/or hepatic injuries, such as ballooning, are present, with or without fibrosis [69]. Recent observations indicate that MASLD can progress to fibrosis even in the absence of MASH, although fibrosis typically arises from MASH. In patients with MASLD, fibrosis may progress by one stage approximately every 14.3 years, whereas in patients with MASH, the progression rate is faster, with fibrosis advancing by one stage every 7.1 years [70].

### 4.2. Second Step: From MASH to Cirrhosis

MASH can lead to the development of cirrhosis, characterized by fibrosis and nodular formation due to constant liver injury stimuli [71], eventually progressing to liver failure or cancer [69]. Recent research has documented that approximately 22% of patients with MASH and advanced fibrosis may progress to compensated cirrhosis in about 2 years. Among these patients with asymptomatic compensated cirrhosis, 19% may develop complications such as ascites, encephalopathy, and esophageal varices within the next approximately 2 years, leading to decompensated cirrhosis [72]. This progression is attributed to chronic hemodynamic changes, systemic inflammation, portal hypertension, and metabolic factors such as diabetes, obesity, dyslipidemia, and the persistence of the primary etiology, in this case, MASLD, thereby increasing mortality in patients. The transition from compensated to decompensated cirrhosis occurs at a rate of 5–7 per year [73].

On the other hand, fibrosis in patients with cirrhosis due to MASH can undergo regression [74], which is associated with lifestyle changes and pharmacological treatment aimed at improving metabolic disturbances, including steatosis and inflammation, which are crucial in the pathogenesis of MASLD [75]. Additionally, controlling factors that are necessary to maintain stellate cell activation is essential [76].

Recently, the term “recompensation” of decompensated cirrhosis has been included, referring to the state in which patients do not present characteristic acute events for more than a year [77]. This recompensation is reflected in the improvement of clinical parameters such as albumin, total proteins, hemoglobin, percentage of basophils, neutrophil-lymphocyte ratio, alanine aminotransferase, and the presence of diabetes. Although further investigation is needed on recompensation in patients with cirrhosis specifically caused by MASLD, the inclusion of this new term demonstrates the need to further investigate the change in how we view the natural history of MASLD as traditionally known and to consider cirrhosis as a bidirectional process that can culminate in regression to fibrosis or progress to decompensation and/or result in the development of HCC [78], a process that is represented in Figure 1.

### 4.3. Third Step: From Cirrhosis to HCC

Moreover, cirrhosis can progress to HCC, associated with multiple causes, including chronic inflammation that damages DNA through ROS [79,80]. These mechanisms lead to the accumulation of genetic mutations and alterations in methylation, which can even occur in patients with mild fibrosis from MASH and result in HCC [81]. Although HCC is not exclusive to patients with MASLD and cirrhosis, it has been observed that patients without the latter condition can develop HCC but are diagnosed at more advanced stages. This is possibly due to insufficient attention given to MASLD in non-cirrhotic stages and its risk for developing HCC [82,83], highlighting the importance of MASLD prevention, as not only advanced stages are associated with fatal outcomes.

There are no ideal molecular markers that allow us to distinguish between cirrhosis and HCC. Studies using diet-induced models tend to fail due to the long time required to develop HCC, and when genetically modified or carcinogenically applied, they are often complicated to translate [84,85]. However, there seem to be two theories that postulate the pathological progression. The first theory assumes the liver with cirrhosis as a pre-cancer stage, and as proliferation and cellular mutations progress, HCC develops. The second theory suggests that cirrhosis-affected liver cells alter hepatocyte proliferation, making them more sensitive to external carcinogenic factors, and since the cellular reproduction rate does not allow DNA repair, accumulated mutations lead to HCC. Among the differentially expressed transcription factors in HCC tissue are BUB1B, NUSAP1, TTK, HMMR, CCNA1, KIF2C [85], TCF4, RUNX1, HINFP, KDM2b, MAF, JUN, NR5A2, NFYA, and AR [86] as candidates for prognostic markers. The adaptive and innate immune systems, being critically involved in tumor development, correlate with the appearance of HCC, with alterations in CD8+ T cells and NKT cells, or, alternatively, even with altered bile acid metabolism [84]. Table 1 describes the estimated evolution time and clinical manifestations of each stage from MASLD to HCC.

## 5. Diagnosis

Liver biopsy acts as the Gold Standard for the definitive diagnosis of steatosis and fibrosis, allowing for the identification of disease severity. The first percutaneous liver biopsy is attributed to Paul Ehrlich in 1883 in Germany [87]. However, this procedure is invasive, painful, and expensive, with potential risks of complications and inaccurate results [88].

The sample obtained in a biopsy represents 1/50,000 parts of the liver. It is necessary to be performed by an experienced operator for accurate interpretation [89]; however, it has been reported that the number of complications does not significantly decrease compared to an inexperienced operator [88], which can lead only to under- or overestimation in interpreting the degree of fibrosis in some cases [90].

Currently, hepatic ultrasound (US) is considered the first-line imaging method for detecting MASLD due to its non-invasive nature, availability, low cost, and absence of radiation [91]. Additionally, it provides complementary data that assist in establishing or ruling out differential diagnoses of other possible hepatobiliary pathologies [92].

On ultrasound imaging, the presence of steatosis is characterized by increased hepatic echogenicity surpassing that of the kidneys, along with reduced clarity of the gallbladder wall, intrahepatic vessels, and diaphragm. The sensitivity of ultrasound in detecting steatosis ranges from 60% to 94%, with specificity varying between 84% and 95% [93].

A meta-analysis of 49 studies evaluating the sensitivity and specificity of hepatic US in detecting moderate to severe MASLD compared to liver biopsy showed that the US had a sensitivity of 84.8% and specificity of 93.6% [94]. A sub-analysis of this study concludes that while the US is highly sensitive for diagnosing moderate to severe steatosis, its sensitivity decreases to 53–66% and specificity to 77–93% in cases of steatosis >5% and <30% [91]. It has also been reported that the US, without the use of elastography, may suggest the presence of cirrhosis by establishing the irregularity and nodularity of the hepatic parenchyma [95].

For the non-invasive determination of MASLD, in addition to the US, techniques based on magnetic resonance imaging (MRI), such as spectroscopy or proton density fat fraction (PDFF) measurement, can be used, which are the most accurate for diagnosing MASLD [96]. The determination of hepatic steatosis grade by MRI has a significant disadvantage in terms of availability [97]. Therefore, MRI spectroscopy is predominantly utilized in research studies.

In patients with risk factors for developing MASLD, such as visceral adiposity, T2DM, or MetS, it is recommended to initiate a diagnostic algorithm consistent with abdominal ultrasound. The determination of liver enzyme levels has low sensitivity and specificity for the diagnosis of MASLD, as it has been observed that these levels can be normal in patients with advanced fibrosis. However, MASLD is often associated with elevated levels of alanine aminotransferase (ALT) [98].

Prediction scales have the advantage of being non-invasive and pose no risk to patients. The most used tests in clinical practice for the diagnosis of MASLD are the fatty liver index (FLI) and the hepatic steatosis index (HSI). These tests use quite common variables (BMI, waist circumference, AST, ALT, GGT, triglycerides, and the presence of T2DM) and are currently proposed as an alternative by several experts [8,99]. Also, the AST to platelet ratio index (APRI), the fibrosis-4 (FIB-4) score, the hepamet fibrosis score (HFS), and the NAFLD fibrosis score (NFS) have a good correlation with the ultrasound-based transient elastography (areas under the ROC curves of 0.79, 0.80, 0.70, and 0.68, respectively) [72,100].

The ultrasound-based transient elastography, or FibroScan, is an option approved by the Food and Drug Administration (FDA) to assist the physician with the liver stiffness measurement (LSM) reported in kilopascals (kPa); it can vary depending on the range of the disease [101,102]. This value correlates with the degree of fibrosis. The estimation is performed using an ultrasonic attenuation wave at 3.5 MHz, and the parameter used to estimate the amount of accumulated fat is known as the controlled attenuation parameter (CAP) and is reported in decibels per meter (dB/m) [103]. However, Eddowes et al. identified a significant error margin in the population with obesity due to the use of probes that are not suitable for the depth between skin and organs. Therefore, XL probes have been developed to assess patients with a greater amount of adipose tissue [104].

In terms of the interpretation of this study, the degree of steatosis is defined according to the number of affected hepatocytes: S0 (<5%), S1 (5–33%), S2 (34–66%), S3 (>66%) [105]. Table 2 shows the cutoff points proposed by Petroff et al. [106].

On the other hand, the LSM can range between 2.5 and 75 kPa [101], and the accurate interpretation also depends on two parameters: the interquartile range (IQR), which reflects the variability of valid measurements and should not exceed 30% [107], and the success rate (the ratio of successful measurements to the total number of acquisitions), which should be at least 60% [101]. The degree of fibrosis is classified with the METAVIR scoring system, with levels oscillating from F0 to F4 [89]. Table 3 shows the actual cutoffs of the LSM proposed by Eddowes et al. [104].

## 6. Complications

MASLD has been shown in various studies to be independently associated with obesity, in particular central adiposity, T2DM, high triglycerides, low HDL cholesterol, and insulin resistance, as well as the presence of MetS [108]. Clinical and epidemiological evidence has reported that MASLD is not only associated with high morbidity and mortality but also with an increase in coronary heart disease, alterations in cardiac function and structure (left ventricular dysfunction, hypertrophy, and heart failure), coronary valve disease (aortic valve sclerosis), and arrhythmias (atrial fibrillation) [109]. In addition, studies in the Mexican population showed that 17% of patients with steatohepatitis will develop cirrhosis [110], which leads to the conclusion that the increase in metabolic comorbidities over the last decades might have been the main driver of the leading cirrhosis etiology.

Despite existing evidence of the implications of living with MASLD and its impact on life expectancy and quality of life, there is a need for greater public health efforts to raise awareness, even among healthcare professionals. A clear example of this was observed in a survey conducted among Mexican gastroenterology fellows, where 22% were unaware of the global prevalence of MASLD, and 29% did not consider it a potentially dangerous disease [111].

## 7. Discussion

This review is in agreement with other recent ones, where no specific pathophysiological mechanisms have been identified to elucidate the progression from MASLD to HCC. However, based on what is known about the accumulated and progressive alterations in individuals with poor lifestyle habits (i.e., overeating, sedentary behavior, recurrent alcohol consumption), it is more than enough to confirm that this trigger is what leads the population to experience this natural progression from MASLD to HCC [112,113]. Excessive alcohol consumption remains an important constant for the development or acceleration of this pathological condition, especially when combined with a poor lifestyle and a state of severe obesity [113].

Although obesity is more prevalent in women in Mexico, DALYs related to high BMI are higher in men [6], indicating a greater risk of mortality and poor quality of life in men due to higher adiposity. This phenomenon is explained by the hormonal and anatomical differences between the sexes, as men have a smaller subcutaneous fat reservoir [25] and an enzymatic and hormonal activity that makes fat storage more difficult [26], placing them at greater risk. Regarding MASLD, although not all individuals with obesity develop it, the risk increases with body weight [10], and its most severe complication, MASH, has a high prevalence in Latin America, especially in Mexico [19]. The lack of infrastructure in the Mexican healthcare system complicates the treatment of obesity and its complications [114], making it urgent to implement effective strategies that promote real lifestyle changes in the population. Moreover, the increase in MASH cases and its relationship with the risk of HCC highlights the need for timely diagnoses and treatments to prevent an increase in mortality from this cause.

Insulin resistance is usually attributed to an increased consumption of carbohydrates, which in turn promotes hyperglycemic states, elevating insulin levels and viciously leading the individual to develop this resistance. However, it is important to understand that it is not only the abnormally frequent stimulation of insulin that will lead to peripheral resistance, but any increase in energy intake above the individual’s requirements, regardless of the macronutrient source, will cause lipotoxicity. This is consistently demonstrated with high-fat diets to stimulate it [35,39,40,41].

It is interesting to note that MASLD has repercussions on the hepatic parenchyma, making it dysfunctional, i.e., leading to fibrosis, without the need for immune cell infiltration or proinflammatory cytokines, which should raise awareness of the need for prompt action even in the presence of minimal hepatic steatosis [70]. On the other hand, it is also necessary to understand that even in a cirrhosis stage, not everything is lost, as compensation should be the goal to pursue. Extremes, like everything in life, are harmful, and MASLD should not be underestimated, as even without cirrhosis, it can predispose to the direct development of HCC [81,82,83], but neither should we condemn a patient with cirrhosis [77,78].

Although liver biopsy remains the gold standard for diagnosing hepatic steatosis and fibrosis, its invasive nature and limitations in accuracy have driven the use of non-invasive methods such as ultrasound, elastography, and magnetic resonance imaging [87,88]. These approaches offer advantages in terms of accessibility, safety, and cost, but each presents limitations in sensitivity and specificity, especially in the early stages of the disease [88,89,90]. Predictive models based on biomarkers and clinical features have also become valuable tools for the initial diagnosis and follow-up of MASLD. Together, a multidimensional diagnostic approach that combines these methods could improve the accuracy and safety of evaluating metabolic liver disease.

Despite the growing evidence of the serious implications of MASLD on morbidity, mortality, and quality of life, there is still a lack of awareness, even among healthcare professionals, highlighting the urgent need to intensify public health efforts to improve education and prevention of this disease [111].

## 8. Conclusions

MASLD is a pathological condition underestimated not only by healthcare professionals but also by official bodies responsible for classifying and addressing such issues, which, over time, can lead to high morbidity and mortality rates and poor quality of life. Lack of understanding of its etiology and pathophysiology, along with current limited diagnostic capabilities, makes the appropriate and timely treatment a challenge. Public health policies aimed at raising awareness, preventing MASLD progression through proper screening, and implementing strategies to improve lifestyle in the population are necessary.

## Figures and Tables

**Figure 1 medicina-61-00088-f001:**
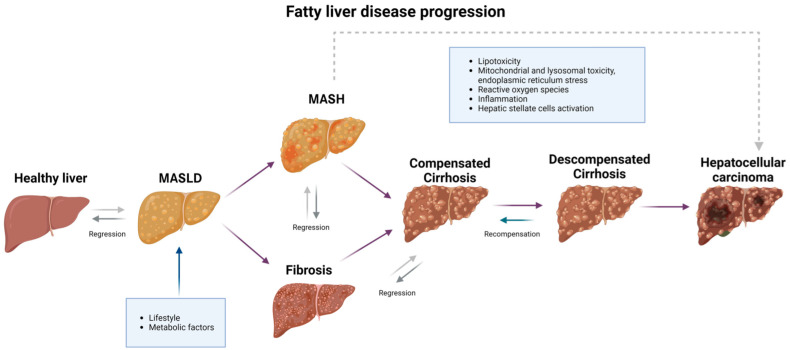
Natural History of MASLD and factors contributing to the progression from a healthy liver to potential outcomes such as hepatocellular carcinoma.

**Table 1 medicina-61-00088-t001:** Natural history of MASLD to HCC, progression, and clinical manifestations.

Liver Stage	Evolution Time for Develop	Clinical Manifestations	References
MASLD	Not determined in this review. The degree of metabolic damage and lipotoxicity may accelerate its onset.	Necroinflammation and metabolic alterations perpetuate a vicious cycle, triggering subsequent stages.	[57,65]
MASH	Not determined in this review. The degree of inflammation and lipotoxicity may accelerate its onset.	Inflammation and/or liver lesions ballooning, with or without fibrosis.	[69]
Liver fibrosis	Progression from MASLD, one stage, every 14.3 years; from MASH, one stage, every 7.1 years.	Usually asymptomatic, although liver morphology continues to evolve due to chronic inflammation.	[70]
Compensated cirrhosis	Progression from MASH and advanced fibrosis in 22% of patients, 2 years.	Usually asymptomatic. Ascites, encephalopathy, and esophageal varices develop during the transition to decompensated cirrhosis.	[72]
Decompensated cirrhosis	Progression from compensated cirrhosis in 19% of patients, 2 years.	Chronic hemodynamic changes, systemic inflammation, portal hypertension, and metabolic alterations.	[72]
Hepatocellular carcinoma	Not determined in this review. The development time depends on the previous stage and can occur from MASLD, although it is less likely.	Genetic mutations and chronic inflammation damage DNA via ROS.	[79,80,81]

**Table 2 medicina-61-00088-t002:** Ranges of CAP Score and Steatosis Grade.

Steatosis Grade	CAP Score (dB/m)
S0	<294
S1	294–309
S2	310–330
S3	≥331

S0: No liver steatosis; S1: mild steatosis; S2: moderate steatosis; S3: severe steatosis.

**Table 3 medicina-61-00088-t003:** LSM Cutoff Points.

Fibrosis Level	LSM Cutoff (kPa)
F0–F1	<8.1
F2	8.2–9.6
F3	9.7–13.5
F4	≥13.6

F0: no fibrosis; F1: fibrous portal expansion; F2: periportal septae; F3: portal-central septae; F4: cirrhosis.

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
