# Peer review of "Natural History of Metabolic Dysfunction-Associated Steatotic Liver Disease: From Metabolic Syndrome to Hepatocellular Carcinoma"

_medicina, 2025, doi:10.3390/medicina61010088_

Round 1
Reviewer 1 Report (Previous Reviewer 1)
Comments and Suggestions for Authors
The authors responded to all my comments and recommendations.
Author Response
Thank you.
Reviewer 2 Report (Previous Reviewer 2)
Comments and Suggestions for Authors
It is a decent review article but the article needs to be improved
The following suggestions are offered:
1. The available data on this subject should be reflected in a tabular form
2. The discussion needs to be augmented. The following maybe discussed and referenced:
a. Lekakis V, Papatheodoridis GV. Natural history of metabolic dysfunction-associated steatotic liver disease. Eur J Intern Med. 2024 Apr;122:3-10. doi: 10.1016/j.ejim.2023.11.005. Epub 2023 Nov 7. PMID: 37940495.
b. Hagström H, Shang Y, Hegmar H, Nasr P. Natural history and progression of metabolic dysfunction-associated steatotic liver disease. Lancet Gastroenterol Hepatol. 2024 Oct;9(10):944-956. doi: 10.1016/S2468-1253(24)00193-6. PMID: 39243773.
Author Response
Comments 1: The available data on this subject should be reflected in a tabular form.
Answer: We appreciate your kind comments, and we answer that the pathophysiology stages are now in a tabular form as you suggest.
Comments 2: The discussion needs to be augmented. The following maybe discussed and referenced.
Answer: The 2 reviews now are discussed in the "discussion" section.
Reviewer 3 Report (Previous Reviewer 3)
Comments and Suggestions for Authors
Thanks to the authors who certainly made a great effort to readjust their previous review to the new MASLD classification and nomenclature. This was necessary to give due weight to their work.
In my opinion, there is still some confusion in the introduction of the classification, which is partly resolved in the later parts of the work.
However, I suggest giving an introductory overview of the classification so that the reader can better understand which entity we are talking about:
in particular, steatotic liver disease is divided into two major entities, MASLD which can evolve into MASH (as specified by the authors) and MASLD associated with increased alcohol consumption (MetALD). I suggest further clarification as to which of these two we are dealing with.
Author Response
Comments 1: In particular, steatotic liver disease is divided into two major entities, MASLD which can evolve into MASH (as specified by the authors) and MASLD associated with increased alcohol consumption (MetALD). I suggest further clarification as to which of these two we are dealing with.
Response 1: We appreciate your kind suggestion, and we gladly say that in the introduction, there is now an explanation about the types of steatotic liver and their definitions (MASLD, MetALD and ALD).
Round 2
Reviewer 2 Report (Previous Reviewer 2)
Comments and Suggestions for Authors
All issues addressed
This manuscript is a resubmission of an earlier submission. The following is a list of the peer review reports and author responses from that submission.
Round 1
Reviewer 1 Report
Comments and Suggestions for Authors
The Natural History of Non-Alcoholic Fatty Liver Disease: From Metabolic Syndrome to Hepatocellular Carcinoma is a review of an important liver disease. As the authors rightly point out, NAFLD can progress to stetohepatitis, cirrhosis, and even hepatocellular carcinoma. There is no doubt that NAFLD is associated with various metabolic diseases, which often occur in people with normal BMI.
In their study, the authors substantiated the need to study NAFLD and analyzed the epidemiology of this disease among the adult population.
The authors also provide data on the etiology and pathogenesis of NAFLD, focusing on the progression from NAFLD to HCC, аnd then they present modern diagnostic criteria, paying attention to complications.
The article is written competently, contains modern data, which is explained by a good selection of modern and relevant sources. The topic has been analyzed sufficiently fully.
All this allowed the authors to formulate a logical and complete conclusion.
The illustration given in the article does not seem very successful to me. It could be made more informative.
It also seems to me that the authors need to point out the connection between the development of liver diseases, including NAFLD, and light pollution, which is one of the factors causing some diseases.
Yue F, Xia K, Wei L, Xing L, Wu S, Shi Y, Lam SM, Shui G, Xiang X, Russell R, Zhang D. Effects of constant light exposure on sphingolipidomics and progression of NASH in high-fat-fed rats. J Gastroenterol Hepatol. 2020 Nov;35(11):1978-1989. doi: 10.1111/jgh.15005. Epub 2020 Feb 23. PMID: 32027419.
Wei L, Yue F, Xing L, Wu S, Shi Y, Li J, Xiang X, Lam SM, Shui G, Russell R, Zhang D. Constant Light Exposure Alters Gut Microbiota and Promotes the Progression of Steatohepatitis in High Fat Diet Rats. Front Microbiol. 2020 Aug 21;11:1975. doi: 10.3389/fmicb.2020.01975. PMID: 32973715; PMCID: PMC7472380.
Areshidze DA, Kozlova MA, Makartseva LA, Chernov IA, Sinelnikov MY, Kirillov YA. Influence of constant lightning on liver health: an experimental study. Environ Sci Pollut Res Int. 2022 Nov;29(55):83686-83697. doi: 10.1007/s11356-022-21655-3.
Abdraboh ME, El-Missiry MA, Othman AI, Taha AN, Elhamed DSA, Amer ME. Constant light exposure and/or pinealectomy increases susceptibility to trichloroethylene-induced hepatotoxicity and liver cancer in male mice. Environ Sci Pollut Res Int. 2022 Aug;29(40):60371-60384. doi: 10.1007/s11356-022-19976-4. Epub 2022 Apr 14. PMID: 35419691; PMCID: PMC9427929.
Chojnacki C, Walecka-Kapica E, Romanowski M, Chojnacki J, Klupinska G. Protective role of melatonin in liver damage. Curr Pharm Des. 2014;20(30):4828-33. doi: 10.2174/1381612819666131119102155. PMID: 24251675.
However, my comments are advisory in nature and do not detract from the high merits of the work.
Author Response
Comments 1. The illustration given in the article does not seem very successful to me. It could be made more informative.
It also seems to me that the authors need to point out the connection between the development of liver diseases, including NAFLD, and light pollution, which is one of the factors causing some diseases.
Response:
Hello, we are grateful for your review and we tell you that the pathophysiology section was redesigned to give the reader a better understanding and also better addressed the causes that lead to HCC. In addition, you are told that the nomenclature of the figure was corrected.
Reviewer 2 Report
Comments and Suggestions for Authors
The review discusses a very common topic with enough data on this subject.
The following are suggested to improve the manuscript:
1. The pathophysiology should be explained better with a diagram/sketch for better comprehension
2. The data on each stage and contributing to the natural history should be depicted in a tabular manner
3. The discussion needs improvement. There are enough references which maybe discussed and added after a better literature review
Comments on the Quality of English LanguageThe language is largely acceptable
Author Response
Comments 1. The pathophysiology should be explained better with a diagram/sketch for better comprehension.
Response:
Hello, we consider that this point is addressed in the following way: the pathophysiology is divided and the information from each stage is concentrated without mixing information in other stages, for a greater understanding of the progress of each of these.
Comments 2. The data on each stage and contributing to the natural history should be depicted in a tabular manner.
Response:
We address this point in conjunction with the first comment, giving greater structure to the evolution of MASLD, in addition to adding more information about the pathophysiology, specifically the transition between cirrhosis to HCC.
Comments 3. The discussion needs improvement. There are enough references which maybe discussed and added after a better literature review.
Response:
We agree, and we added a discussion section where we address the most important points that we wish to convey in this research.
Reviewer 3 Report
Comments and Suggestions for Authors
Congratulations to the authors for this very thorough review of NAFLD, but I think it is necessary to focus on the new definition of MAFLD. As already highlighted by other studies, the NAFLD definition probably includes individuals with a lower risk of disease progression, whereas MAFLD identifies a more homogeneous group of patients with fatty liver associated with metabolic dysfunction. Therefore, I suggest that the authors revise their review and focus their remarks on MAFLD rather than NAFLD, so that their considerations and conclusions are more robust, minimising the confounding factors.
Author Response
Comments 1. I think it is necessary to focus on the new definition of MAFLD. As already highlighted by other studies, the NAFLD definition probably includes individuals with a lower risk of disease progression, whereas MAFLD identifies a more homogeneous group of patients with fatty liver associated with metabolic dysfunction. Therefore, I suggest that the authors revise their review and focus their remarks on MAFLD rather than NAFLD, so that their considerations and conclusions are more robust, minimising the confounding factors.
Response:
Hello, indeed, now we have concentrated the terminology in the new proposal by Rinnella et al., 2023, I hope you find the change appropriate, we even changed the figure so that it respects the new terminology.